# First Belgian Report of Ertapenem Resistance in an ST11 *Klebsiella Pneumoniae* Strain Isolated from a Dog Carrying *bla_SCO-1_* and *bla_DHA-1_* Combined with Permeability Defects

**DOI:** 10.3390/antibiotics11091253

**Published:** 2022-09-15

**Authors:** Hanne Debergh, Margo Maex, Cristina Garcia-Graells, Cécile Boland, Marc Saulmont, Koenraad Van Hoorde, Claude Saegerman

**Affiliations:** 1Service Foodborne Pathogens, Sciensano, B-1050 Brussels, Belgium; 2Veterinary Bacteriology Service, Sciensano, B-1050 Brussels, Belgium; 3Research Unit in Epidemiology and Risk Analysis Applied to Veterinary Sciences (UREAR-ULiège), Fundamental and Applied Research for Animal and Health (FARAH) Center, University of Liège, B-4000 Liège, Belgium; 4Service Bacterial Diseases, Sciensano, B-1050 Brussels, Belgium; 5Association Régionale de Santé et d’Identification Animales, B-5590 Ciney, Belgium

**Keywords:** *Klebsiella pneumoniae*, sequence type 11, companion animal, sepsis, antimicrobial resistance (AMR), carbapenem, emerging risk, *OmpK37*, hybrid sequencing

## Abstract

*Klebsiella pneumoniae* of sequence type (ST) 11 is a hyper-epidemic nosocomial clone, which is spreading worldwide among humans and emerging in pets. This is the first report, to the best of our knowledge, of multidrug-resistant (MDR) *K. pneumoniae* ST11 carrying *bla_SCO-1_* and *bla_DHA-1_*, isolated from a four-month-old dog in Belgium. Antimicrobial susceptibility testing (AST) of the isolate, performed via broth microdilution following the European Committee on Antimicrobial Susceptibility Testing (EUCAST) guidelines, revealed resistance to eight different classes of antimicrobials, including carbapenems, in particular ertapenem, third-generation cephalosporins and fluoroquinolones. A hybrid approach, combining long- and short-read sequencing, was employed for in silico plasmid characterization, multi-locus sequence typing (MLST) and the identification and localization of antimicrobial resistance (AMR) and virulence-associated genes. Three plasmids were reconstructed from the whole-genome sequence (WGS) data: the conjugative IncFIB(K), the non-mobilizable IncR and the mobilizable but unconjugative ColRNAI. The IncFIB(K) plasmid carried the *bla_SCO-1_* gene, whereas IncR carried *bla_DHA-1_*, both alongside several other antimicrobial resistance genes (ARGs). No virulence genes could be detected. Here, we suggest that the resistance to ertapenem associated with susceptibility to imipenem and meropenem in *K. pneumoniae* could be related to the presence of *bla_SCO-1_* and *bla_DHA-1_*, combined with permeability defects caused by point mutations in an outer membrane porin (*OmpK37*). The presence of the *bla_SCO-1_* gene on a conjugative IncFIB(K) plasmid is worrisome as it can increase the risk of transmission to humans, to animals and to the environment.

## 1. Introduction

The opportunistic bacterium *Klebsiella pneumoniae* is a common source of antimicrobial-resistant hospital-acquired infection and the increase in antimicrobial resistance (AMR) in general is one of the greatest global public health challenges of this time [1]. The six leading pathogens for deaths attributed to AMR, i.e., *Escherichia coli*, *Staphylococcus aureus*, *K. pneumoniae, Streptococcus pneumoniae*, *Acinetobacter baumannii* and *Pseudomonas aeruginosa,* caused an estimated 929,000 (660,000–1,270,000) deaths in 2019 [2].

The World Health Organization (WHO) has included *Enterobacteriaceae* resistant to third-generation cephalosporins and/or carbapenems in its list of urgent threats, with each causing 50,000–100,000 deaths in 2019 [2,3]. *K. pneumoniae* is naturally resistant to penicillin and is a known trafficker of acquired resistance genes to multiple antimicrobials, such as to β-lactams and carbapenems [1,4]. Multidrug resistance (MDR), which is defined as resistance to more than three antimicrobial classes, in addition to resistance to ampicillin for *K. pneumoniae*, has evolved many times, leading to hundreds of distinct lineages [1]. Many of them will persist locally, but a subset of lineages have evolved into global problem clones with wide dissemination worldwide [1,5]. These include, amongst others, the well-studied clonal groups CG258, CG101 (CG43), CG15 and CG307 [1,5]. CG258 includes sequence types (STs) ST11, ST258 and ST512, which have often been reported as resistant to third-generation cephalosporins and carbapenems [1]. Different mechanisms of resistance against carbapenems exist and carbapenem resistance may be caused by the production of carbapenemases, such as *bla_KPC_*, *bla_VIM_*, *bla_NDM_* and *bla_OXA-48_* [1]. On the other hand, the co-occurrence of alterations in the expression or lesions of outer membrane porins (OMPs), combined with the production of β-lactamases that possess sparse carbapenemase activity, may also lead to reduced carbapenem susceptibility, most often to ertapenem [6].

There are limited data on clones found in non-human niches; however, some overlap is seen between clinical isolates and other sources. Global problem clones, such as ST15 and ST11, are increasingly being reported in companion animals [7,8,9]. In humans, carbapenem-resistant *K. pneumoniae* (CRKp) may lead to serious infections such as urinary tract infections (UTIs), pneumonia and septicemia, which could be incurable and fatal due to limited therapeutic options [10,11]. In companion animals, *K. pneumoniae* has been reported to cause UTIs, upper respiratory tract infections, pyometra and bloodstream infections (septicemia) [12,13]. Although carbapenems are not the standard of care in companion animals, infections with carbapenem-resistant *K. pneumoniae* are increasingly being reported [14]. Due to the close contact between companion animals and humans, they may constitute an important reservoir of carbapenem-resistant *Enterobacterales* (CRE), as such raising a public health concern [14,15,16]. Therefore, it is crucial to understand the molecular mechanisms involved in this resistance. In this study, we aimed to investigate the genotypic mechanisms related to the ertapenem resistance in a *K. pneumoniae* isolate from a companion animal with fatal sepsis.

## 2. Results and Discussion

### 2.1. Phenotypic Resistance Testing

*K. pneumoniae* strain Kpn1695 was screened for ESBL production and carbapenem resistance using McConkey + CTX (1 mg/L) and CHROMID*^®^* CARBA SMART Agar. Growth was observed on McConkey + CTX and on the CARB side of CHROMID*^®^* CARBA SMART Agar. Further phenotypic resistance testing was performed via broth microdilution.

*K. pneumoniae* strain Kpn1695 was resistant to ampicillin (minimal inhibitory concentration (MIC) > 32 mg/L), azithromycin (MIC > 64 mg/L), cefepime (MIC = 2 mg/L), cefotaxime (MIC = 8 mg/L), cefotaxime/clavulanic acid (MIC = 8 mg/L), cefoxitin (MIC > 64 mg/L), ceftazidime (MIC > 8 mg/L), ceftazidime/clavulanic acid (MIC = 32 mg/L), chloramphenicol (MIC = 64 mg/L), ciprofloxacin (MIC > 8 mg/L), ertapenem (MIC = 0.25 mg/L), gentamicin (MIC > 16 mg/L), nalidixic acid (MIC > 64 mg/L), sulfamethoxazole (MIC > 512 mg/L), temocillin (MIC = 16 mg/L), tetracycline (MIC > 32 mg/L) and trimethoprim (MIC = 8 mg/L). However, this strain was susceptible to amikacin (MIC ≤ 4 mg/L), colistin (MIC ≤ 1 mg/L), imipenem (MIC = 0.25 mg/L), meropenem (MIC = 0.06 mg/L) and tigecycline (MIC = 1 mg/L) (Table 1).

Epidemiological cutoff (ECOFF) values are used to distinguish microorganisms with (non-wild-type) and without (wild-type) phenotypically detectable acquired resistance genes to a certain antibiotic agent. EUCAST does not provide ECOFFs for all antibiotics included in the Sensititre™ plates EUVSEC 3 and EUVSEC 2 (Appendix A
Table A1 and Table A2). In the absence of ECOFF for *K. pneumoniae*, those for *E. coli*, the indicator bacterium for *Enterobacteriaceae*, were applied (Table A1).

### 2.2. Genotypic Characterization

By performing in silico multi-locus sequence typing (MLST) determinations, we classified *K. pneumoniae* Kpn1695 as ST11 (allelic profile: gapA, 3; infB, 3; mdh, 1; pgi, 1; phoE, 1; rpoB, 1; and tonB, 4). This sequence type was one of the most frequently detected sequence types related to third-generation cephalosporin resistance and/or CRKp in the Europe-wide EuSCAPE study [1]. The clinical high-risk ST11 clones, which are part of the clonal group (CG) 258, have become global problem strains due to their wide dissemination and their multiple drug resistance (MDR) characteristics [1]. ST11, which is derived from ST258 by recombination, represents about 12% of CRKp in Europe and is much more widely distributed than other STs from CG258, namely, ST258 and ST512 [1,17]. In China, ST11 represents the single dominant cause of CRKp human infections [1]. It has also been isolated from non-human niches such as from poultry in China [18] and from dogs and cats in Spain and Switzerland [7,19].

Moreover, the polysaccharide capsule of *K. pneumoniae* plays a major role in its virulence and can, with over 138 distinct combinations identified to date [1,20,21], be used as an epidemiological marker [20]. This strain was typed as a KL105, a clade which has been described previously in clinical ST11 *K. pneumoniae* strains isolated in Europe and Latin America [22,23]. Furthermore, this ST11 KL105 clade has successfully disseminated in Europe, bearing variable β-lactamases. It is, however, most often related to ESBL strains or *bla**_DHA-1_* producers with the presence of different plasmid types [22]. Genotypic characterization identified Kpn1695 as O2v2 (Table 2). Currently, 12 distinct O-loci have been described [21,24]. Clinical strains are most often typed as serotypes O1 and O2 [21]. No clear association has been established between the K-locus or O-locus and niche or host specialization [25].

### 2.3. Plasmid Analysis

A total of 27 antibiotic resistance genes (ARGs) were found in Kpn1695 (Table 2, Figure 1 and Figure 2, Appendix A
Figure A1). Following hybrid assembly from long and short reads, analysis of the assembly with ABRicate and PlasmidFinder revealed the presence of three plasmids: IncFIB(K), IncR and ColRNAI [26,27]. A nucleotide BLAST search of the plasmid sequences against the GenBank database showed the highest similarity for the IncFIB(K) plasmid with a *K. pneumoniae* isolate bearing plasmid tig00000000 (GenBank accession no. CP021713.1) with 81% query coverage and 99.78% identity. A query coverage of 100% and 99.97% identity was observed for the IncR type plasmid with pMBR_DHA-1_1CO (GenBank accession no. CP049718.1). This *K. pneumoniae* strain was isolated from the veterinary setting in Switzerland. The ColRNAI type plasmid displayed 100% query coverage and 99.38% identity with pC17KP0039-3 (GenBank accession no. CP052402.1), which was retrieved from a human *K. pneumoniae* sample.

The IncFIB(K) plasmid, named pKPn1695_SCO-1, is a multidrug resistance conjugative plasmid (length = 192,683 bp) carrying the following resistance genes: the CARB-type β-lactamase *bla**_SCO-1_* gene, *tet*(D), *bla_SHV-2_*, *aac(3)IIa*, *bla**_TEM-1A_*, *sul1*, *aadA1*, *qnrE1* and *emrE* (Figure 1). The IncFIB(K)-type plasmid has previously been described in clinical *K. pneumoniae* ST11 from humans and is amongst the most commonly detected plasmid replicon types in *K. pneumoniae* [28,29]. This plasmid replicon type has recently been identified in *K. pneumoniae* isolates belonging to several STs from non-human sources, such as from bovine mastitis and in raw milk [30], in pets [19] and in pigs [31]. The *emrE* gene, encoding a multidrug exporter, is associated with resistance to a wide range of toxic cationic hydrophobic compounds, such as the disinfectant quaternary ammonium compounds (QACs). It is a known disinfectant resistance gene and belongs to the small multidrug resistance (SMR) family. This β-lactamase has been detected in *Pseudomonas aeruginosa*, *Acinetobacter baumannii* and *Vibrio cholera* but occurs in low frequencies in *Enterobacteriaceae* [32]. It has been described before in *E. coli* and *K. pneumoniae* [32,33]. The conjugative plasmid pKPn1695_SCO-1 carries the β-lactamase *bla**_SCO-1_* gene (Figure 1), which has previously been described as having carbenicillinase activity [32].

A circular plasmid diagram was generated using BRIG and the *bla_SCO-1_*-carrying plasmid was compared to a selection of its top BLAST hits: plasmid tig00000000 (strain AR_0129, accession nr. CP021713), pBK13043_1 (accession nr. CP020838) and plasmid tig00000002 (strain AR_0115, accession nr. CP020072) (Figure 1). Strikingly, high similarity was seen between these plasmids in the backbone, but little coverage was seen in the MDR region and *bla_SCO-1_* was not present in any of these plasmids. The lack of homology in the MDR region of pKPn1695_SCO-1 to its top BLAST hits suggests transmission of the MDR region from other plasmids. It is likely that this occurred in several recombination events through insertion sequences (IS), as several IS were observed flanking the MDR region without similarity to the other plasmids. The *bla_SCO-1_* gene on pKPn1695_SCO-1 displayed the highest BLAST similarity to a *bla_SCO-1_* gene originating from *Acinetobacter baumannii* (accession nr. EF063111), which might suggest a recombination event. The fact that this gene is lying on a conjugative plasmid highlights the risk of its spread to other *K. pneumoniae* strains or other potential hazardous species.

The non-mobilizable IncR plasmid (length = 56,132 bp) named pKpn1695_DHA-1 carried *bla**_DHA-1_*, *bla**_OXA-1_, aac(6′)Ib-cr5*, *aph(3′)-Ia, qnrB4, mph(A), catB3, arr3, sul1* and *emrE* (Figure 2). IncR plasmids have been previously linked to the spread of *bla**_DHA-1_* in ST11 *K. pneumoniae* strains [34]. The *bla**_DHA-1_* cephalosporinase is currently reported worldwide in *K. pneumoniae*. The *bla**_DHA-1_* β-lactamase is usually co-expressed with plenty of other ARGs, such as extended-spectrum β-lactamases (*bla**_CTX-M_*_-_, *bla**_SHV_*-types), oxacillinases (*bla**_OXA-1_*), penicillinases (bla*_TEM_*-type), carbapenemases (*bla**_OXA-48_*, *bla**_KPC-2_*), aminoglycosides (*aacA, aadA, armA*), fluoroquinolones (*qnrB4, aac6**′-1b-cr*) and sulfonamide (*sul1*) resistance genes, of which some are also present in the current isolate Kpn1695 (Table 2) [35]. The presence of *bla**_DHA-1_* β-lactamase in combination with OmpK35/OmpK36 porin loss has been linked to carbapenem resistance in *K. pneumoniae* before [36].

The mobilizable but unconjugative plasmid ColRNAI did not carry ARGs.

Upon comparing phenotypic properties with genotypic data, a complete match was observed for this isolate, except for rifampicin and fosfomycin. No phenotypic data were collected for these antibiotics as they are not included in the EUCAST panel for the AMR monitoring of zoonotic and indicator bacteria, following the European Decision (EUVSEC 3 and EUVSEC 2) [37]. However, *arr-3* and *fosA* were detected by genotypic means and are known to confer resistance to rifampicin and fosfomycin, respectively (Table 2 and Figure A1). On the contrary, phenotypic resistance was observed for trimethoprim without the presence of known resistance genes.

Phenotypic screening tests on CHROMID*^®^* CARBA SMART Agar showed growth on the CARB side of the plate. The latter is usually indicative of the presence of *bla**_KPC_* and bla*_NDM-1_* carbapenemase genes. However, no phenotypic resistance was seen for imipenem and meropenem and, similarly, whole-genome sequencing did not identify the presence of such genes. Growth on the CARB side of the selective plate could be explained by the presence of the *bla**_SCO-1_* gene, which has weak carbenicillinase activity [32]. Phenotypic screening tests that show growth on the CARB side of CARBASMART and resistance to ertapenem but susceptibility to imipenem and meropenem could be used for epidemiological purposes to screen for the presence of the *bla**_SCO-1_* gene in other *Enterobacteriaceae*. Further investigations are required to verify this hypothesis by screening *Enterobacteriaceae* bearing the *bla_SCO_**_-1_* gene for growth on CARBASMART and phenotypic resistance to ertapenem.

An increasing number of studies have reported a role for *AmpC* enzymes, such as *bla_DHA-1_*, in carbapenem resistance in clinical ST11 *K. pneumoniae* strains [36]. All of them report a combination of *AmpC* cephalosporinase expression and outer membrane porin gene alterations, such as in *OmpK35*/*OmpK36* [36,39]. In contrast to an earlier finding linking mutations in *OmpK37* to lower carbapenem MICs [40], we observed elevated MIC values for ertapenem. Furthermore, plasmid pKpn1695_DHA-1 contains the gene *ampR*, an upstream regulatory gene governing *bla_DHA-1_* synthesis [41]. This gene is also involved in the expression of virulence factors in *Pseudomonas aeruginosa* [41] and plays a pleiotropic role in the pathogenic process of *K. pneumoniae* since it is involved in biofilm formation and type 3 fimbrial gene expression [41]. The loss of function of *ampR* has been linked to decreased MIC values of β-lactams, whereas the presence of a functional *ampR* gene showed high resistance rates to β-lactams and higher β-lactam activity. Nakano et al. suggested that the higher β-lactamase activity of *bla_CFE-1_*, a plasmid-encoded AmpC β-lactamase, depends on the functioning of *ampR*, rather than on its own high hydrolyzing activity [42]. More research is needed to examine whether *ampR* can play a similar role in regulating *bla_DHA-1_* expression and thus affect β-lactam and carbapenem resistance.

No virulence factors were found using Kleborate [24].

The recovery of an ST11 strain from a companion animal carrying several ARGs of critical importance on a conjugative plasmid is worrisome as this could potentially be transferred to other animals, its owner or other members of the family, or to the environment (excreta), thus threatening public health. The dog could potentially have obtained the strain from its owner or from other sources. However, we could not obtain a sample from the owner to confirm this. A correlation of ARGs between dogs and their owners has already been demonstrated [43].

### 2.4. Chromosome Analysis

Using the CARD database, a match was found for the outer membrane protein *OmpK37*, located on the genome (Figure A1), which had 18 SNPs and 43 holes and showed 95.50% coverage compared to accession nr. AJ011502.1. An in-depth analysis of the outer membrane protein revealed twelve synonymous mutations, six missense mutations and one in-frame insertion. Reduced carbapenem susceptibility in the presence of *bla_SCO-1_*, combined with porin deficiency related to outer membrane porins *OmpK35/OmpK36*, has been previously described [33,36]. Furthermore, two chromosomal mutations contributing to antimicrobial resistance to quinolones, gyrA-83L and parC-80I, were detected.

In the same database, a match was observed for a*rnT* and *eptAB*, which are related to chromosomal polymyxin resistance. In Gram-negative bacteria, polymyxin resistance usually occurs via the addition of positively charged moieties (e.g., pEtN and 4-amino-4-deoxy-L-arabinose (L-Ara4N)) to the lipid A component of LPS [44]. These lipid A modifications alter the net negative charge with a decreased electrostatic interaction to polymyxins as a result. Lipid A modifications in pEtN can arise from either a chromosomally encoded pEtN transferase gene, *eptAB* or a plasmid-encoded *mcr* gene. In contrast, L-Ara4N modification of lipid A is activated by a transferase encoded by *arnT* that is exclusively encoded on the chromosome [45,46]. However, phenotypic data did not show resistance against colistin (Table 1). Protein analysis of *arnT* showed an amino acid similarity of 99.82% and 100% query coverage to *arnT* from *K. pneumoniae* (accession nr. FO834906.1) and *eptB* showed 99.30% amino acid similarity and 100% query coverage to a *K. pneumoniae* entry in the GenBank protein database with accession nr. FO203501.1. On the other hand, *phoP* showed 100% amino acid similarity with 95% query coverage to a *Klebsiella* spp. entry in the GenBank reference sequence protein database with accession nr. WP_004151175.1 and *phoQ* showed an amino acid similarity of 100% and 100% query coverage to *phoQ* from *K. pneumoniae* (accession nr. WP_004147969.1). Further studies could help to decipher if such variants in *arnT* and *eptAB* could affect the functionality of the genes, and if the expression of *phoP* and *phoQ* could be altered, resulting in colistin susceptibility in *K. pneumoniae* Kpn1695.

## 3. Materials and Methods

### 3.1. Bacterial Isolate

A four-month-old female dog with bacteremia was admitted to a veterinary clinic in Belgium in June 2020, where it died of fatal sepsis. Post-mortem biopsies of the liver and brain were individually inoculated on both Gassner (Oxoid, Basingstoke, UK) and Columbia media (Oxoid, UK) and were incubated at 37 °C ± 2 °C for 24 h in aerobic conditions and 24 h in aerobic conditions at 37 °C with 5% CO_2_, respectively. Suspected *K. pneumoniae* colonies were purified on Columbia agar (Oxoid, UK) and species confirmation was performed via MALDI-TOF mass spectroscopy (Bruker Daltonics, Bremen, Germany). The isolate identified as *K. pneumoniae* (strain Kpn1695) was screened for ESBL production and carbapenem resistance using McConkey + CTX (1 mg/L, Biorad, Hercules, CA, USA) and CHROMID^®^ CARBA SMART Agar (Biomérieux, Marcy-l’Étoile, Frankrijk), respectively [14,37].

### 3.2. Phenotypic Resistance Testing

*K. pneumoniae* Kpn1695 was subjected to antimicrobial susceptibility testing using the broth microdilution method, according to ISO 20776-1:2019, using EUVSEC3 and EUVSEC2 Sensititre^TM^ plates (Trek Diagnostic Systems; Thermo Scientific, Waltham, MA, USA). *E. coli* ATCC 25922 was used for quality control. The antibiotic panel EUVSEC3 Sensititre^TM^ plate included amikacin, ampicillin, azithromycin, cefotaxime, ceftazidime, chloramphenicol, ciprofloxacin, colistin, gentamicin, meropenem, nalidixic acid, sulphamethoxazole, tetracycline, tigecycline and trimethoprim. The antibiotic panel EUVSEC2 Sensititre^TM^ plate included cefepime, cefotaxime, cefotaxime/clavulanic acid, cefoxitin, ceftazidime, ceftazidime/clavulanic acid, ertapenem, imipenem, meropenem and temocillin.

Results were interpreted according to the European Committee on Antimicrobial Susceptibility Testing (EUCAST) guidelines, using epidemiological cutoff values (ECOFF) as described in the Commission Implementing Decision 2020/1729 [37]. In the absence of epidemiological cutoff values for *K. pneumoniae*, those for *E. coli*, the indicator *Enterobacteriaceae* bacterium, were applied (Table A1 and Table A2).

### 3.3. Genotypic Resistance Testing Using Whole-Genome Sequencing (WGS)

Genomic DNA of *K. pneumoniae* Kpn1695 was extracted from 10 mL of pure culture using the genomic Tip 20/G kit (Qiagen, Benelux B.V., Venlo, The Netherlands), following the manufacturer’s instructions. The purity of the DNA was evaluated with a Nanodrop 2000 spectrophotometer (ThermoFisher Scientific, Waltham, MA, USA) and DNA was determined with a Qubit 3.0 fluorometer (Thermo Fisher Scientific, Waltham, MA, USA). A Nextera XT DNA sample preparation kit (Illumina, San Diego, CA, USA) was used for library preparation and sequencing was performed on an Illumina MiSeq instrument (Illumina, San Diego, CA, USA) using MiSeq V3 chemistry, as described by the manufacturer’s protocol, for the production of 2 × 300 bp paired-end reads. Raw MiSeq sequencing reads were trimmed using Trimmomatic (v0.38) with the ‘SLIDINGWINDOW:4:20′ option [47]. The leading and trailing bases of a read were removed when the Phred dropped below a score of 3 [47]. The trimmed MiSeq reads were used in a de novo assembly generated with SPAdes v3.15.4 [48]. Raw sequencing data were submitted to NCBI Genbank under BioProject PRJNA870711, with accession nrs. CP103301, CP103302, CP103303 and CP103304.

Long-read library preparation was carried out using the Rapid Barcoding Sequencing kit (SQK-RBK004), according to the standard protocol provided by the manufacturer, Oxford Nanopore Technologies (ONT, Oxford, UK). The constructed library was loaded onto an R9.4.1 MinION flow cell (FLO-MIN106) and sequenced for 48 h on a MinION Mk1C. Base-calling and demultiplexing were performed with Guppy (v6.0.1). Subsequently, long reads were checked with NanoStat and NanoPlot (v1.32.0) in order to gain a general overview of the read quality. The adapter and barcode sequences were removed with Qcat (v1.1.0). Low-quality (<q8) and short (<10,000 bp) reads were removed with NanoFilt (v2.0.0) [49]. Complete genome sequences were obtained using a hybrid de novo assembly with Unicycler (v0.4.7) with default settings [50]. Short-read assembly and hybrid assembly data were annotated using Prokka and manually curated (version 1.14.6) [26].

Acquired antibiotic resistance-encoding gene prediction was carried out using NCBI AMRFinder (version 3.10.18), ResFinder v4.0 [51], ARG-ANNOT [52] and CARD, using default settings [53]. Mobile genetic elements were analyzed with MobileElementFinder using default settings [54]. Analyses of the plasmid replicon type were performed using PlasmidFinder 1.3 [27]. BIGSdb was used for the determination of the sequence type. The K locus, O locus and the presence of virulence factors were analyzed using Kleborate with default settings (Version v2.0.4) [55]. To determine the position of the acquired resistance genes and/or mutations on either chromosome or plasmid, the four contigs obtained in the hybrid assembly were analyzed separately with ABRicate with default settings (v1.0.1). PLASCAD was used to locate genes associated with transfer. Additional annotation of plasmid sequences was performed using PROKKA version 1.14.6 [26]. Protein analysis was performed with BLASTP with the RefSeq Select protein database. Visualization of in-depth plasmid analysis was carried out using the BLAST Ring Image Generator (BRIG) [38].

## 4. Conclusions

In this work, we investigated the molecular mechanism causing ertapenem resistance in a *K. pneumoniae* strain isolated from a four-month-old dog using hybrid sequencing analysis. We described ertapenem resistance that could be related to the presence of *bla**_SCO-1_*** and *bla**_DHA-1_***, combined with porin lesions in the outer membrane porin *OmpK37*. This is the first report describing the combination of *bla**_SCO-1_*** and *bla**_DHA-1_*** in *K. pneumoniae.* An in-depth plasmid analysis, carried out with the use of hybrid sequencing technologies, showed the presence of the *bla**_SCO-1_*** gene on a conjugative IncFIB(K) plasmid, possibly increasing the risk of its transmission to humans, to animals and to the environment. Our research highlights the importance of combining long- and short-read sequencing in AMR research as this approach overcomes common challenges involved in plasmid reconstruction. The investigation of other *Enterobacteriaceae* that also harbor *bla**_SCO-1_***-carrying plasmids through phenotypic screening tests with growth on the CARB side of CARBASMART and phenotypic resistance to ertapenem but susceptibility to imipenem and meropenem could be of epidemiological importance, as the extent of their spread in veterinary and clinical isolates is not fully understood.

## Figures and Tables

**Figure 1 antibiotics-11-01253-f001:**
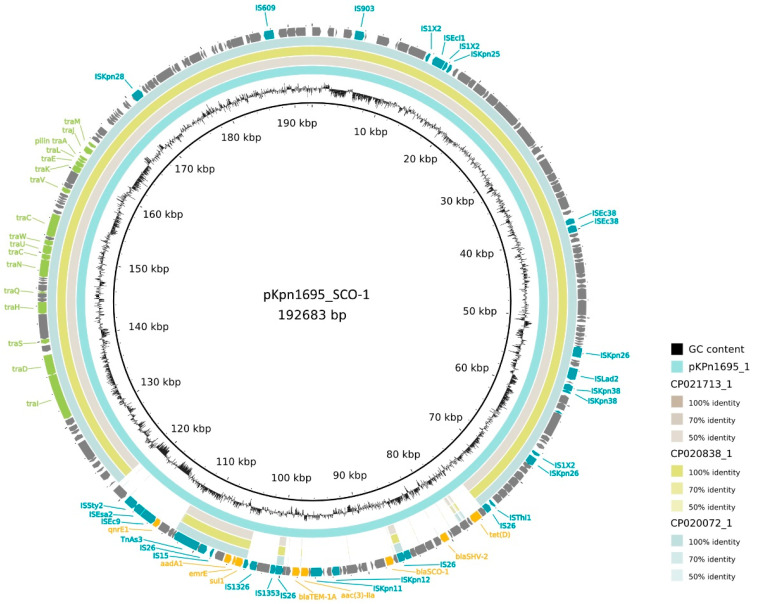
Genomic map and BRIG comparison of the *bla_SCO-1_*-carrying conjugative IncFIB(K) plasmid and three closely genetically related plasmids retrieved from GenBank. The inner light-blue ring represents pKpn1695_SCO-1 and the gray (*K. pneumoniae* of unknown origin), green (clinical *K. pneumoniae*) and light blue (*K. pneumoniae* of unknown origin) rings represent three plasmids with high overall sequence similarities (accession nrs. CP021713_1, CP020838_1 and CP020072_1). The outer ring depicts antimicrobial resistance genes in yellow, insertion sequences in blue and transfer genes in green. The circular map was generated with BLAST Ring Image Generator (BRIG) software [38].

**Figure 2 antibiotics-11-01253-f002:**
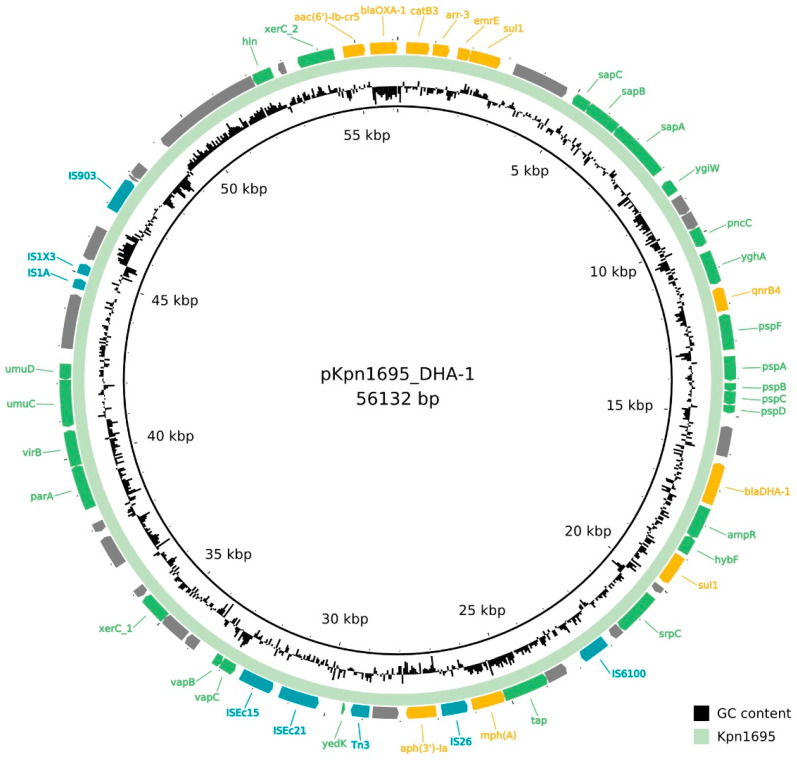
Circular map of the *bla_DHA-1_*-carrying IncR plasmid. The green ring represents pKpn1695_DHA-1. The outer ring indicates antimicrobial resistance genes in yellow, insertion sequences in blue and green genes with various functions. The circular map was generated with BLAST Ring Image Generator (BRIG) software [38].

**Table 1 antibiotics-11-01253-t001:** Minimal inhibitory concentrations (MIC, mg/L) of Kpn1695 using Sensititre™ plates EUVSEC 3 and EUVSEC 2.

EUVSEC 3		EUVSEC 2	
Antibiotic	MIC (mg/L)	Antibiotic	MIC (mg/L)
Amikacin	≤4	Cefepime *	2 **
Ampicillin *	>32 **	Cefotaxime	8 **
Azithromycin *	>64 **	Cefotaxime/clavulanic acid *	8 **
Cefotaxime	>4 **	Cefoxitin	>64 **
Ceftazidime	>8 **	Ceftazidime	8 **
Chloramphenicol *	64 **	Ceftazidime/clavulanic acid *	32 **
Ciprofloxacin	>8 **	Ertapenem *	0.25 **
Colistin	≤1	Imipenem	0.25
Gentamicin	>16 **	Meropenem	0.03
Meropenem	0.06	Temocillin	16 **
Nalidixic Acid *	>64 **		
Sulfamethoxazole *	>512 **		
Tetracycline	>32 **		
Tigecycline	1		
Trimethoprim *	8 **		

ECOFF values of *K. pneumoniae* or *E. Coli* (*) (in the absence of ECOFF for *K. pneumoniae*) were used to interpret the obtained MIC values. Resistance is shown by ‘**’.

**Table 2 antibiotics-11-01253-t002:** Phenotypic antimicrobial susceptibility testing via the broth microdilution method and genotypic characterization through whole-genome sequencing of *K. pneumoniae* Kpn1695.

Parameter	Characterization
MLST ^1^	11
K locus	KL 105
O locus	O2V2
IncFIB(K)	β-lactams (*bla_SCO-1_*, *bla_TEM-1_*, *bla_SHV-2_*), Aminoglycosides (*aac(3)-IIa, aadA1)* Sulfonamide (*sul1*) (Fluoro)quinolones (*qnrE1*) Tetracycline (t*et(D))*
IncR	β-lactams *(bla_DHA-1_*, *bla_OXA-1_*) Aminoglycosides (*aac(6′)-Ib-cr5*, *aph(3′)-Ia*) Phenicols (*catB3*) Sulfonamide (*sul1*) Rifamycin (*arr-3*) * Quinolones (*qnrB4)* Macrolide (*mph(A))*
ColRNAI	No antibiotic resistance genes found
Chromosome	Fosfomycin (*fosA6*) * Quinolones (GyrA-83L, ParC-80I) Polymyxins (*arnT, eptAB, phoPQ*) ** Macrolide (*mdfA*) Phenicol *(oqxA, oqxB)*
	β-lactams (*bla_SHV-11_*) Outer membrane porins (*OmpK37*, *OmpA*)

^1^ MLST = multi locus sequence typing, * no phenotypic data available to test for phenotypic rifamycin resistance, ** no phenotypic resistance was observed for colistin.

## Data Availability

The data that support the findings of this study are available from the corresponding author upon request. Raw sequencing data were submitted to NCBI Genbank under BioProject PRJNA870711.

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
