# Peer review of "First Belgian Report of Ertapenem Resistance in an ST11 Klebsiella Pneumoniae Strain Isolated from a Dog Carrying blaSCO-1 and blaDHA-1 Combined with Permeability Defects"

_antibiotics, 2022, doi:10.3390/antibiotics11091253_

Round 1
Reviewer 1 Report
The article describes in a very explicit manner the molecular mechanism causing ertapenem resistance in 358 a K. pneumoniae strain isolated from a four-month old dog. It is very interesting to read and the possible explanations given by the authors are supported by the research. There are some minor modifications in my opinion that need to be made but only related to the presentation of the information and not the actual content. The authors should revise the text as there is some typo errors and also the names of the genes to be uniformly written (all italics).
Author Response
Comment #1: The article describes in a very explicit manner the molecular mechanism causing ertapenem resistance in 358 a K. pneumoniae strain isolated from a four-month old dog. It is very interesting to read and the possible explanations given by the authors are supported by the research. There are some minor modifications in my opinion that need to be made but only related to the presentation of the information and not the actual content. The authors should revise the text as there is some typo errors and also the names of the genes to be uniformly written (all italics).
Response #1:
We thank Reviewer #1 for the positive appraisal of the proposed paper. As requested, all names of the genes are now in italics.
Response #2:
The authors have revised the text and some typo’s:
Line 28: IncFIB was modified to IncFIB(K).
Line 59: KPC, VIM, NDM, OXA-48 was modified into: blaKPC, blaVIM, blaNDM and blaOXA-48
Line 115: “clonal group 258 (CG258)” was modified into “CG258” as it was written in full in line 112
Line 142: the typo in “99.387%” was modified into “99.38%”
Line 200: “**” was added in the legend of table 2 before “no phenotypic resistance was observed for colistin.”
Line 283: The typo in eptB was changed into eptAB
Line 284: the discussion over the findings of phoP and phoQ was missing in this sentence. We completed it as follows: “ Further studies could help to decipher if such variants in arnT and eptAB could affect the functionality of the genes, and if the expression of phoP and phoQ could be altered, resulting in colistin susceptibility in K. pneumoniae Kpn1695.”
Line 397: “interpre-tation” was modified into “interpretation”
Line 401: “inter-pre-tation” was modified into “interpretation”
Reviewer 2 Report
The manuscript shares interesting findings about ertapenem resistance and molecular mechanism behind it. Hybrid sequencing data even showed the presence of genes that might lead to easy transmission and spread/rise of MDR bugs. To my understanding the manuscript is scientifically sound and well written
Author Response
Comment #1:
The manuscript shares interesting findings about ertapenem resistance and molecular mechanism behind it. Hybrid sequencing data even showed the presence of genes that might lead to easy transmission and spread/rise of MDR bugs. To my understanding the manuscript is scientifically sound and well written
Response:
We thank Reviewer #2 for the positive comments on the proposed paper.
Reviewer 3 Report
Thank you for submitting this paper investigating the molecular mechanism causing ertapenem resistance in a K. pneumoniae strain isolated from a dog using hybrid sequencing analysis. The article is of interest to researchers in the field. There are, however, corrections required to the text as follows:
line 19 “MDR” & line 64” CRKp”& line 131 “ARGs”: please write in full at the first mention
lines 19-20 “isolated from a companion animal in Belgium” : please write the isolation source “a four-month-old dog in Belgium”
line 26 “virulence-associated genes. .”: delete the second full stop
lines 96& 97: please delete “Error reference source not found”
line 107 “and” not italic
lines 145- 148 “The IncFIB(K) type plasmid has previously been described in clinical K. pneumoniae ST11 from humans and is amongst the most detected plasmid replicon types in K. pneumoniae [28,29].”: It is preferred to add that “this plasmid replicon type has recently been identified in K. pneumoniae isolates from bovine mastitis and raw milk (https://doi.org/10.3389/fmicb.2021.770813).
lines 182: write the abbreviation only “ARGs” as it should be written in full at line 131.
line 193: broth microdilution instead of “micro broth dilution”
Author Response
Response to Reviewer 3 Comments
Comment #1:
line 19 “MDR” & line 64” CRKp”& line 131 “ARGs”: please write in full at the first mention
Response #1
line 19: “MDR” was changed to multi-drug resistant (MDR) in the abstract.
Response #2:
Line 64: “CRKp” was changed into carbapenem-resistant K. pneumoniae (CRKp). At line 110 only the abbreviation CRKp was used.
Response #3:
Line 131(became line 132): “ARGs” was changed into antibiotic resistance genes (ARGs)
Comment #2:
lines 19-20 “isolated from a companion animal in Belgium” : please write the isolation source “a four-month-old dog in Belgium”
Response:
Line 19-20: The sentence was modified to “This is the first report, to the best of our knowledge, of a multi-drug resistant (MDR) K. pneumoniae ST11 carrying blaSCO-1 and blaDHA-1 isolated from a four month-old dog in Belgium.”
Comment #3:
line 26 “virulence-associated genes. .”: delete the second full stop
Response:
line 26: The second full stop after “virulence-associated genes” was deleted.
Comment #4:
lines 96& 97: please delete “Error reference source not found”
Response:
Lines 96&97: The sentence “Error reference source not found” was not present in the word document to revise so no adaptations were made.
Comment #5:
line 107 “and” not italic
Response:
Line 107: the font of “and” was adapted to not italic.
Comment #6:
lines 145- 148 “The IncFIB(K) type plasmid has previously been described in clinical K. pneumoniae ST11 from humans and is amongst the most detected plasmid replicon types in K. pneumoniae [28,29].”: It is preferred to add that “this plasmid replicon type has recently been identified in K. pneumoniae isolates from bovine mastitis and raw milk (https://doi.org/10.3389/fmicb.2021.770813).
Response:
Lines 145-148 (became line 147- 149): This sentence was added which includes the suggested reference: “This plasmid replicon type has recently been identified in K. pneumoniae isolates belonging to several STs from non-human sources, such as from bovine mastitis and in raw milk [30], in pets [19] and in pigs [31].”
Comment #7:
lines 182: write the abbreviation only “ARGs” as it should be written in full at line 131.
Response #1:
Line 182 (became line 185): “The mobilizable but unconjugative plasmid ColRNAI did not carry antibiotic resistance genes (ARGs).“ was modified into “ The mobilizable but unconjugative plasmid ColRNAI did not carry ARGs.”
Response #2
The same modification was done for line 247
Comment #8:
line 193: broth microdilution instead of “micro broth dilution”
Response:
Line 193 (became line 196): micro broth dilution was modified to broth microdilution